# Kidney Fibrosis and Oxidative Stress: From Molecular Pathways to New Pharmacological Opportunities

**DOI:** 10.3390/biom14010137

**Published:** 2024-01-22

**Authors:** Francesco Patera, Leonardo Gatticchi, Barbara Cellini, Davide Chiasserini, Gianpaolo Reboldi

**Affiliations:** 1Division of Nephrology, Azienda Ospedaliera di Perugia, 06132 Perugia, Italy; francesco.patera@ospedale.perugia.it; 2Department of Medicine and Surgery, University of Perugia, 06132 Perugia, Italy; leonardo.gatticchi@unipg.it (L.G.); barbara.cellini@unipg.it (B.C.)

**Keywords:** kidney fibrosis, oxidative stress, mitochondrial energy imbalance, mineralocorticoid signaling, hypoxia-inducible factor, sodium-glucose cotransporter 2

## Abstract

Kidney fibrosis, diffused into the interstitium, vessels, and glomerulus, is the main pathologic feature associated with loss of renal function and chronic kidney disease (CKD). Fibrosis may be triggered in kidney diseases by different genetic and molecular insults. However, several studies have shown that fibrosis can be linked to oxidative stress and mitochondrial dysfunction in CKD. In this review, we will focus on three pathways that link oxidative stress and kidney fibrosis, namely: (i) hyperglycemia and mitochondrial energy imbalance, (ii) the mineralocorticoid signaling pathway, and (iii) the hypoxia-inducible factor (HIF) pathway. We selected these pathways because they are targeted by available medications capable of reducing kidney fibrosis, such as sodium-glucose cotransporter-2 (SGLT2) inhibitors, non-steroidal mineralocorticoid receptor antagonists (MRAs), and HIF-1alpha-prolyl hydroxylase inhibitors. These drugs have shown a reduction in oxidative stress in the kidney and a reduced collagen deposition across different CKD subtypes. However, there is still a long and winding road to a clear understanding of the anti-fibrotic effects of these compounds in humans, due to the inherent practical and ethical difficulties in obtaining sequential kidney biopsies and the lack of specific fibrosis biomarkers measurable in easily accessible matrices like urine. In this narrative review, we will describe these three pathways, their interconnections, and their link to and activity in oxidative stress and kidney fibrosis.

## 1. Introduction

Chronic kidney disease (CKD) affects nearly 8–16% of the population in developed countries [1,2,3]. Due to its increasing incidence, its poor prognosis, and its high medical and social costs, renal fibrosis diffused to the interstitium, the vessels and the glomerulus, is the main histologic characteristic of kidney failure [4,5,6,7,8,9]. Fibrosis can be seen in the final process, common to all kidney diseases independently from their primary cause, and currently it is a key target of new drug treatments. The knowledge of the basic mechanisms involved in kidney fibrosis and its link with oxidative stress is therefore fundamental to understand and exploit the opportunities offered by new treatments.

Mechanistically, the landmark review by Remuzzi et al. [10] described the excessive glomerular filtration of albumin, immunoglobulins, and transferrin as the main events eliciting the production of Angiotensin II (Ang II). In the proximal tubular cells, the action of Ang II induces the expression of the nuclear factor kappa B (NF-κB), a cytoplasmic protein complex that when active, migrates in the nucleus, driving the production of an inflammatory cytokine pattern that includes IL-1, IL-6, IL-8, RANTES, MCP-1, endothelin-1, and transforming growth factor beta 1 (TGF-1β). These molecules cause the migration of inflammatory cells (mainly T cells) into the interstitium and, eventually, the accumulation of collagen. This is an inflammatory pathway triggered by heterogeneous stimuli, explaining why different kidney diseases, such as autoimmune, metabolic, iatrogenic, viral, etc., merge in this common final pattern and subsequently in renal fibrosis. Moreover, proteinuria and excessive intrarenal production of Ang II lead directly to the secretion of Collagen IV and other types of collagen, causing extracellular matrix deposition [10,11]. More recent studies suggest that other factors might be triggered by proteinuria, like the reduced expression of Klotho and the increased expression of Fibroblast Growth Factor 23 (FGF-23). FGF-23 is a peptide that induces the proliferation of fibroblasts, starting the fibrosis cascade [12,13]. The reduction in Klotho seems to be associated with cellular senescence and consequent fibrosis in many tissues. This reflects the ubiquitous action of soluble Klotho (sKlotho). This happens very early and seems to be triggered not only by frank proteinuria, but even by small increments of urinary albumin excretion. This could be the reason why even mild levels of albuminuria are associated with increased cardiovascular (CV) and renal risk independently of renal function [14,15].

Mineralocorticoid receptor (MR) activity plays a pivotal role in the development of fibrosis in the glomeruli, vessels, and interstitium. Several studies showed reduced cardiac, vascular, and renal fibrosis in animal models, corresponding to the absence of MR activity and/or an MR antagonism independently of blood aldosterone levels [16].

Starting from the early 2000s, drugs known as angiotensin-converting enzyme inhibitors (ACEi), angiotensin receptor blockers (ARBs), and mineralocorticoid receptor antagonists (MRAs) have been employed in nephroprotection because they act in blocking the renin–angiotensin–aldosterone system (RAAS) at different levels. However, despite wide availability and evidence-based guidelines and recommendations on RAAS blockers [17], an unacceptable number of patients still progress to kidney failure [1]. Thus, in recent years, enormous efforts have been made in developing new compounds or repurposing already approved medications [18,19] acting at different levels in the kidney and mainly targeting glucose handling and metabolism, the hypoxia-inducible factor–prolyl hydroxylase domain protein (HIF1-PHD) axis, and, more selectively, the RAAS [20]. 

An insightful investigation [21] on the role of oxidative stress in kidney structural damage, especially fibrosis, allowed researchers to identify other pathways as potential targets for the development of new therapeutic approaches. For instance, in a recent review, Dobrek [22] examined the possible use of inhibitors of key proteins involved in oxidative damage (NADPH oxidase, protein kinase C, xanthine oxidase, and transforming growth factor β). Moreover, given the significant role of heme oxygen-ase-1 as a target of the nuclear factor erythroid 2-related factor 2 (Nrf2) antioxidant defense system in CKD, the use of phenolic and non-phenolic phytochemicals as activators of the Nrf2-heme oxygenase-1 signaling pathway were proposed [23]. However, the kidney’s protective actions of the above phytochemicals were investigated only in animal models and not in humans.

Taken together, there is a large and growing number of potentially nephroprotective compounds in active development; however, for practical and clinical reasons, in this review, we will address only medications recently approved, recommended by guidelines, and available for prescription for CKD patients [17]. We will summarize and explore the molecular and clinical scenarios of sodium-glucose cotransporter 2 inhibitors (SGLT-2i), non-steroidal MRAs, and inhibitors of the HIF1-PHD pathway (HIF1-PHI), focusing on their current and future clinical perspectives for kidney fibrosis and their impact on the management of CKD (Figure 1).

## 2. Molecular Scenario

### 2.1. Glucose and Mitochondrial Energy Imbalance in Kidney Disease

Filtered glucose can alter the cellular metabolism of endothelial, glomerular, and proximal tubule (PT) cells and progressively contribute to impair kidney function [24,25].

At the kidney level, excessive glucose load, even if transitory, induces massive and long-lasting cellular modifications, like adaptive hypertrophy, glomerular hyperfiltration, rearrangement of metabolic pathways, and epigenomic changes in kidney cells [26,27,28]. For instance, proximal tubular (PT) cells in the presence of glycosuria require an abnormal amount of energy and oxygen consumption for ATP synthesis to sustain continuous glucose reabsorption into the bloodstream through sodium–glucose cotransporters (SGLTs) [29]. This altered energy-metabolism flux generates an oxidative burst that leads to mitochondrial dysfunction by triggering inflammatory and fibrotic pathways, which ultimately determines the decline of the glomerular filtration rate (GFR) and renal scarring [30,31,32].

At the molecular level, PT cells are forced by the excessive glucose availability to overwhelm its catabolism via the glycolytic and mitochondrial oxidative pathways, inducing the generation of damaging levels of reactive oxygen species (ROS) and depleting intracellular content of oxygen. Under normal conditions, PT cells utilize fatty acid oxidation (FAO) as the preferential source for producing ATP, and the impairment of oxidative phosphorylation (OXPHOS) downstream of FAO is thought to be one of the key drivers of tubular injury and fibrinogenesis [33,34]. Notably, positive regulation of FAO transcriptional regulators (e.g., PPARα and PGC-1α) and mediators (e.g., CPT1α) plays a protective role against severe cellular damage in PT cells [35,36,37]. Moreover, the uptake control of free fatty acids (FFAs) via the inhibition of the free fatty acid transporters CD36 or FATP2 in PT cells permits a reduction in the lipotoxicity produced following an excess of lipid accumulation in cells that are unable to efficiently perform FAO [35,38,39,40]. 

Hyperglycemia-induced ROS overproduction is responsible for a plethora of detrimental effects due to the alteration of the cellular oxidant–antioxidant balance. Indeed, a reduced ROS scavenging ability brings an accumulation of irreversibly modified cellular sugars, lipids, nucleic acids, and proteins that can stimulate secondary metabolisms like the polyol, hexosamine, AGE/RAGE, and protein kinase C (PKC) pathways, which are able to disrupt cellular homeostasis [41,42,43,44,45,46]. These activated pathways induce several downstream signaling events as the activation of an inflammatory response mediated by the nuclear factor-κB (NF-κB) and/or JAK/STAT3 pathways, thus mediating the release of inflammatory cytokines (e.g., TNFα), chemokines, adhesion molecules, and fibrinogenic factors (e.g., TGF-β1) [47,48,49,50]. Additionally, the activity of phosphoinositide 3-kinase (PI3K)/protein kinase B (Akt) exerts a tight control over cell proliferation, differentiation, and apoptotic processes by in turn regulating the glycogen synthase 3 β (GSK3β), mTOR (mammalian target of rapamycin), and Forkhead Box Protein O1 (FoxO1) signaling effectors [51,52,53,54]. Nonetheless, the oxidative stress-induced nuclear translocation of the nuclear factor erythroid-2-related factor 2 (Nrf2) transcription factor can promote the expression of antioxidant enzymes while counteracting NF-κB activation, which simultaneously ameliorates cellular defense against oxidative damage and dampens the resulting inflammatory response [55,56,57]. The combination of an augmented inflammatory response, which is responsible for the activation of immune cells and fibroblast-mediated collagen deposition, and a hypertrophic expansion of renal cells altogether contributes to tubular atrophy and glomerular hyperfiltration accompanied by persistent proteinuria and, in the end, loss of kidney functionality.

On the other hand, PT cells of patients with diabetic kidney disease (DKD) exhibit aberrant oxygen consumption that creates a cellular environment characterized by a low oxygen tension and ATP content. Under these conditions, oxygen and energy substrate deprivation can promote the activation of hypoxia-inducible factor 1 α (HIF1α) and of AMP-activated protein kinase (AMPK), which are responsive for oxygen and AMP levels, respectively. HIF-1α is transcription factor that can regulate the expression of the glycolytic enzymes hexokinase-1 (HK1) and phosphofructokinase (PFKL) to promote ATP production with anerobic glycolysis instead of aerobic mitochondrial OXPHOS [58]. AMPK is a modulator of mTOR, a serine/threonine kinase that mediates cell growth control based on the cellular nutrient status [59]. Both HIF-1α and AMPK pathways contribute to fuel the uncontrolled tissue growth and injury accumulation in the PT cells of diabetic individuals with CKD [59].

Therefore, the preservation of mitochondrial function is crucial for the prevention of kidney failure, and it could be achieved by limiting the glucose reuptake mediated by SGLT transporters. The use of SGLT inhibitors represents a potential therapeutic strategy to prevent the establishment of metabolic anomalies, oxidative damage, inflammatory bursts, and hypoxic environments, which are at the basis of renal fibrotic tissue formation and drive DKD progression/severity, at early stages. 

### 2.2. Glucagon-like Peptide-1 in Kidney Disease

Glucagon-like peptide-1 (GLP-1) is a peptide hormone that plays a critical role in glucose homeostasis and insulin secretion [60]. It is produced from the pro-glucagon gene (*GCG*), which is subjected to post-translation processing by prohormone convertase enzymes (PCSKs). PCSK1, expressed mainly in intestinal cells and in the brain, cleaves the pro-hormone in GLP-1, while PCSK2 produces a longer peptide that includes both GLP-1 and GLP-2 [60]. Therefore, the production of pro-glucagon peptides is regulated by the tissue-specific expression of proteases, and the intestine and brain are the main organs producing GLP-1 [60]. The effects of GLP-1 are mediated by the interaction the GLP-1 receptor (GLP-1R) that is expressed at high levels in the pancreas [61]. Activation of GLP-1R by GLP-1 induces pleiotropic effects on glucose metabolism, promoting insulin secretion and reducing blood glucose levels. In parallel, GLP-1R activation inhibits the release of glucagon from pancreatic alpha cells, helping the suppression of hepatic glucose production and contributing to the overall reduction in blood glucose levels [61]. The metabolic effects of GLP-1 are not restricted to glucose metabolism but may influence other physiological districts, like the cardiovascular system and the kidneys. Indeed, GLP-1 can stimulate vasodilation via the induction of nitric oxide (NO) production [62], decreasing platelet activation and inflammation and contributing to overall cardiovascular health [62]. In the kidneys, GLP-1 seems to have important functions related to renal protection. GLP-1R is expressed in the kidneys across different cell types [63], and its expression is lower in DKD patients [64]. The nephroprotective effect of GLP-1 does not seem to correlate with glucose lowering. In an animal model of diabetic nephropathy, GLP-1R agonists were able to reduce proteinuria and ameliorate glomerular filtration without modifying blood pressure or body weight [65]. In humans, GLP-1 infusion can increase sodium excretion and improve the glomerular filtration rate [66,67].

### 2.3. Mineralocorticoids in Kidney Disease

Aldosterone acts on Na^+^/K^+^-ATPase to promote sodium reabsorption, potassium excretion, and water retention, thus leading to hypertension, but it also mediates some untoward pro-inflammatory and pro-fibrotic effects in the kidneys [68]. It is known that in diabetes, hyperglycemia can increase Ang II production, as well as upregulate MR synthesis, thus further worsening tissue damage [69]. Notably, local aldosterone synthesis promotes renal fibrosis by stimulating fibroblast proliferation through growth factor activation, as well as fibronectin production, thus confirming how the hyperactivation of the RAAS system plays a key role on the pathogenesis of DKD. 

The action of aldosterone depends on the activation of a nuclear receptor called the mineralocorticoid receptor (MR), and the expression of 11β-hydroxysteroid dehydrogenase type 2 in the cells of the distal nephron is crucial to confer specificity for aldosterone over glucocorticoids [68]. In the resting state, MR localizes in the cell cytosol, bound to heat-shock proteins. Upon aldosterone binding, it translocates to the nucleus, where it binds the hormone-responsive elements to initiate the transcription of target genes. Preclinical data indicate that the molecular effects of MR activation in kidney disease are related to (i) oxidative stress, due to increased ROS production through the induction of NADPH oxidases and/or decreased ROS detoxification through repression of glucose-6-phosphate dehydrogenase; (ii) inflammation, due to increased leucocyte infiltration, pro-inflammatory cytokine production, and adhesion molecule production; and (iii) promotion of fibrosis through the increased production of connective tissue growth factors, leading to the proliferation of fibroblasts and the deposition of the extracellular matrix [70]. 

All these finding have prompted the targeting of MR using MR antagonists (MRAs). The first MRA approved was spironolactone, but its reduced specificity led to the development of the more selective eplerenone, which attenuated renal fibrosis and reduced oxidative stress in preclinical studies [71]. Recently, to improve the efficacy and reduce the side effects of MRAs, new-generation MRAs with a non-steroidal structure have been developed. The most effective is finerenone, which behaves as an inverse agonist and inhibits cofactor recruitment to MRs in the ligand-free state [72]. 

### 2.4. HIF-PHD Axis: Molecular Mechanism and Its Role in Kidney Diseases

Hypoxia-induced factor (HIF) is the master regulator of adaptative responses to reduced oxygen availability [73]. HIF is a heterodimeric protein complex generally composed of two proteins; the first one, HIF-1 alpha (HIF1α), is codified by the *HIF1A* gene, while the second subunit, HIF-1 beta (HIF1β), or aryl hydrocarbon receptor nuclear translocator protein, is codified by the *ARNT* gene [74] and is expressed constitutively. Upon heterodimerization, the complex has transcriptional activity and binds to hypoxia response elements (HREs) within the promoters of many different downstream targets, activating their transcription [73]. Major targets of the HIF complex are genes involved in angiogenesis, red blood cell development, glycolysis, oxidative stress, and other genes involved in oxygen sensing and metabolic conditioning to hypoxia [75]. 

The HIF complex is regulated in a variety of ways. A major pathway is the PHD–VHL–HIF axis, where, besides HIF, prolyl hydroxylase domain-containing proteins (PHDs) and Von Hippel–Lindau tumor suppressor (VHL or pVHL) are the key proteins regulating HIF activation/deactivation. In normoxia, HIF1α is hydroxylated by PHDs, and particularly by PHD2, an α-ketoglutarate/2-oxoglutarate-dependent dioxygenase that contains Fe^2+^ in the active site. Hydroxylated HIF can interact with VHL, acting as a substrate-recognition subunit of the Cullin-RING E3 ubiquitin ligase complex and therefore targeting HIF for proteasomal degradation [76]. In hypoxia, hydroxylation is inhibited, thereby decreasing HIF degradation and promoting the formation of the heterodimeric complex HIF1α/HIF1β that can activate transcription of target genes in the nucleus [73]. 

HIF is also regulated by another dioxygenase called factor-inhibiting HIF-1 (FIH1). FIH1 can hydroxylate an asparagine residue on HIF1α, preventing interaction with the p300 and CBP proteins which act as coactivators of the transcriptional heterodimeric complex, eventually inhibiting the transcription of hypoxia-related genes [77]. FIH1 has more affinity for O_2_ when compared to members of the PHD family; therefore, it is still active when oxygen levels are very low [78]. Combined regulation by PHD proteins and FIH1 can provide a dynamic response to hypoxia within a wide range of oxygen tensions.

In the kidney, hypoxia represents one of the main consequences of fibrosis, since expansion of fibrotic tissue is associated with the loss of peritubular capillaries, reduced blood flow, and renal anemia [79,80]. In addition, kidneys show low oxygen tension when compared to other organs [81] and display an increase in oxygen demand according to several models mimicking different pathologies with renal involvement, such as diabetes and hypertension [82,83]. These studies underscore the dynamic regulation of oxygen demand in the kidney to supply enough energy for cellular activities such as transport, electrolyte regulation, and several other processes through the aerobic metabolism [84].

Three isoforms of HIFα are expressed in the kidney: HIF-1α, HIF-2α, and HIF-3α. Studies have mainly investigated the roles of HIF-1α and HIF-2α in the kidney. The two isoforms seem to have a different pattern of expression in kidney cells, where HIF-1α is enriched in tubular cells (TECs), while HIF-2α is mainly expressed in endothelial cells and interstitial fibroblasts [85]. However, recent single-cell atlas data have also shown relatively high expression of HIF-2α in TEC cells at the mRNA level [86]. Both isoforms have shown clear associations with renal fibrosis. In a model of cisplatin-induced renal fibrosis, HIF-1α was increased and promoted at the onset of the pathology, activating NOTCH-1 signaling [87]. Other studies have shown that HIF-1α is able to activate a series of pro-fibrotic pathways, including secretion of inflammatory mediators such as IL-1β and TNF-α, induction of epithelial–mesenchymal transition (EMT), and vascular remodeling [88]. HIF-2α is also prominently involved in renal fibrosis. In a unilateral ureteral obstruction (UUO) model of mice, knock-out of HIF-2α attenuated renal fibrosis through a mechanism involving sirtuin-1 (SIRT1) [89]. On the other hand, it has been shown that long-term activation of HIF-2α may alter the course of renal fibrosis, improving renal function and decreasing several EMT markers [90]. 

Xue Li and coworkers [91] showed that in a mouse model of tubular proximal cells, damage induced by folic acid (FA) and relative mitochondrial failure, treatment with the HIF stabilizer Roxadustat prior to FA administration reduced cell apoptosis, stabilized intercellular junctions, and promoted the expression of aquaporin 1 (AQP1) and aquaporin 2 (AQP2). Moreover, interstitial inflammatory cell infiltration after FA injection was reduced as well as collagen deposition. The concentration of HIF-1α in mice treated with Roxadustat was elevated until day 3 from the FA injection and then decreased, while in the control group, it remained elevated until day 7, the time at which the histologic signs of tubular toxicity were evaluated. In the latter group, persistently elevated levels of HIF-1α were related to mitochondria fragmentation and reduction in crest expression, a sign of metabolic suffering. As a consequence, the intracellular levels of ATP decreased in the control group, while the group treated with Roxadustat showed stable ATP levels and decreased ROS levels. A meaningful finding was that in the damaged tubular cells of the non-treated group, the mitochondrial pattern shifted from a fusion pattern to a fission pattern, as demonstrated by the hyperexpression of fission proteins such as FIS 1 e Drp 1 and the lower expression of fusion proteins such as Opa1 and Mfn1. In the Roxadustat group, the pattern shifted to a fusion pattern, ameliorating the aerobic metabolism [91]. 

Taken together, these studies show how regulation of HIF activation and/or expression may directly influence the development of pro-fibrotic pathways, even though contrasting results are often reported and may be associated to different outcomes for renal health [92].

## 3. Clinical Scenario

### 3.1. Sodium Glucose Transporter 2-Inhibitors (SGLT2i)

Sodium glucose transporter inhibitors (SGLT2i) are approved to treat type 2 diabetes (T2DM) in several countries. These drugs reversibly block glucose reabsorption in the S1 and S2 segments of the proximal tubule. Decreasing glucose transport reduces energy needs and therefore oxygen consumption, leading to a reduced production of free-radical species. Reduced metabolic stress improves aerobic metabolism [93] and decreases ROS production and the inflammatory cytokine response, eventually leading to an improvement in kidney fibrosis. Incidentally, the reduction in proximal tubule glucose adsorption brings a massive load of sodium to the distal section and particularly to the macula densa. This activates the tubulo-glomerular feedback that causes the contraction of the afferent arteriola. This is a crucial event because the contraction of the afferent arteriola reduces glomerular pressure, hyperfiltration, proteinuria, and the local production of Ang-II [94]. 

From a clinical evidence perspective, there are three randomized clinical trials (RCTs) specifically designed to test the efficacy of SGLT2i on primary renal outcomes in CKD patients. The main findings of these landmark trials are summarized below. 

The CREDENCE trial (Table 1) randomized 4401 patients with eGFRs between 30 and <90 mL/min/1.73 m^2^ and urinary albumin excretion (UACR) of 300–5000 mg/g on top of the maximum tolerated therapy with ARBs or ACEi. Participants randomized to 100 mg/day canagliflozin vs. the placebo showed a significant 34% reduction in the incidence of primary renal composite outcome, which was defined as end-stage kidney disease requiring dialysis or kidney transplant, a sustained decline of the estimated GFR (eGFR) > 15 mL/min/1.73 m^2^, doubling of serum creatinine (sCr), or death from renal or cardiovascular causes [95]. 

In DAPA-CKD (Table 1), 4304 patients with a mean eGFR of 43.1 mL/min/1.73 m^2^ were randomly assigned to 10 mg/day dapagliflozin or a placebo in addition to the maximum tolerated doses of ACEi or ARBs [96]. Inclusion criteria were an eGFR between 25 and 75 mL/min/1.73 m^2^ and UACR > 200 and <5000 mg/g. Out of 4304 participants, 2906 had diabetic kidney disease (DKD), while 1398 had other CKD diagnoses. The primary endpoint was a decline of at least 50% in the estimated GFR (confirmed by a second serum creatinine measurement after ≥28 days), the onset of end-stage kidney disease, or death from renal or cardiovascular causes. The primary endpoint occurred in 9.2 vs. 14.5% of patients in the dapagliflozin vs. placebo groups, respectively (HR 0.61; 95% CI 0.51–0.72) [97]. 

EMPA-Kidney (Table 1) randomized 6609 CKD patients with a mean eGFR of 37.5 mL/min/1.73 m^2^, who were randomly assigned to 10 mg/day empagliflozin or a placebo [98]. Inclusion criteria were an eGFR of at least 20 but less than 45 mL/min/1.73 m^2^, regardless of the level of albuminuria, or an eGFR of at least 45 but less than 90 mL/min/1.73 m^2^ with a UACR of at least 200 at the screening visit. Out of 6609 participants, 2057 had DKD while 4452 had other CKD diagnoses [98]. The primary endpoint was the first occurrence of kidney disease progression or death from cardiovascular causes. CKD progression was defined as ESKD (the initiation of maintenance dialysis or kidney transplant), a sustained decrease in eGFR to less than 10 mL/min/1.73 m^2^, a sustained decrease of at least 40% from the baseline eGFR, or death from renal causes. The primary endpoint occurred in 13.1% vs. 16.0% patients in the empagliflozin vs. placebo groups, respectively (HR 0.72; 95% CI 0.64–0.82).

Taken together, these recent trials showed a consistently positive and protective effect on the primary renal outcome. Notably, DAPA-CKD and EMPA-KIDNEY showed a highly significant protective effect for non-DKD patients, and in EMPA-KIDNEY, this effect was independent from the albuminuria level [99]. 

These findings prompted international guidelines [17,100] to include SGLT2i among the pillars of the modern approach to treatment of DKD and CKD.

### 3.2. Glucagon-like Peptide-1 Receptor Agonists (GLP-1 RAs)

Glucagon-like receptor agonists (GLP-1 RAs) have been extensively studied in cardiovascular outcome trials (CVOTs) of patients with type 2 diabetes (T2DM) and established or high risks of atherosclerotic cardiovascular disease (ASCVD) either with or without CKD [101]. A comprehensive systematic review of CVOTs [102] clearly showed that GLP-1 RAs reduced the risk of major adverse cardiovascular events (MACEs), all-cause mortality, and worsening kidney function in patients with T2DM, most of whom had ascertained cardiovascular disease, to a significant extent. Thus, based on the available clinical evidence, the American Diabetes Association (ADA) [100] recommended GLP-1 RAs with demonstrated cardiovascular disease benefit as part of a comprehensive cardiorenal risk-reduction strategy for patients with T2DM. 

In a recent network meta-analysis of randomized controlled trials, Shi and coauthors [103] confirmed a significant protective effect of GLP-1 RAs on kidney outcomes (OR 0.83, 95% CI 0.75–0.92), although with moderate certainty of the evidence. According to grading of recommendations, assessment, development, and evaluation (GRADE) approach, the authors rated down the certainty of the evidence to moderate because the composite kidney outcome was heterogeneously reported across the included trials. 

Collectively, the clinical trials and systematic reviews published so far concur in assigning a potential nephroprotective action to GLP-1 RAs, likely due to both indirect (improvement of blood pressure and glucose control, weight loss) and direct (restoration of normal intrarenal hemodynamics, prevention of ischemic and oxidative damage) effects [104]. The encouraging findings about GLP-1 RAs, although not obtained in kidney disease-focused primary-outcome trials, might add an additional therapeutic option and potentially provide an upstream treatment for patients with kidney disease. How these findings on secondary outcomes will extend to kidney disease-focused primary-outcome trials remains to be established and will be important in determining the specific role of GLP-1 RAs for patients with CKD/DKD. To this purpose, new evidence will soon be available. The first dedicated GLP1-RA outcome trial of patients with DKD, the FLOW trial (Effect of semaglutide versus placebo on the progression of renal impairment in people with type 2 diabetes and chronic kidney disease) [105], was stopped early for efficacy on 10 October 2023 [106]. FLOW (Table 1) is a randomized, double-blind, parallel-group, multinational phase IIIb trial designed to evaluate the effect of semaglutide on kidney outcomes in participants with CKD and T2DM (*n* = 3534) originally expected to be completed in late 2024. Hopefully, the FLOW results, expected during the first half year of 2024, will clarify the extent to which semaglutide can reduce the progression of CKD in addition to the established beneficial metabolic and cardiovascular effects.

### 3.3. Non-Steroidal Mineralocorticoid Receptor Antagonists

MR activity is a potent driver for fibrosis, and besides well-known activation pathways, oxidative stress can activate MRs [107]. The relationship between MRs and oxidative stress is bidirectional; in fact, in one way, MR activation provokes reactive oxygen species (ROS) production, and in the other way, ROS activates MRs. 

Activation of MRs contributes to the maintenance of a state of inflammation via NK-kB activation and therefore the production of IL-6, TNF-α, and IFN-Y. Furthermore, MR activation in macrophages can lead to a switch towards an M1 population, a macrophage subset that amplifies chronic inflammatory reactions and produces fibrosis [108,109,110]. 

These pathways have been the basis for the clinical use of MR antagonists (MRAs). Many years ago, the mainstay of the MRA family were spironolactone and eplerenone, two steroidal non-selective MR antagonists [111]. 

Spironolactone showed great efficacy in heart failure (HF) patients. The most important study, the RALES trial, showed a significant difference in mortality (46% vs. 35%; *p* < 0.0001) in the placebo vs. spironolactone group, respectively, in 1663 pts with III–IV NYHA class HF who were followed for 24 months [112]. The positive and significant effect was also demonstrated in the improvement of NYHA class with spironolactone in all the prespecified subgroups. In CKD patients, several studies demonstrated the efficacy of spironolactone in the reduction in proteinuria but none was powered enough to demonstrate a significant reduction in GFR decline [16]. Bianchi and coworkers [113] conducted a controlled study of patients with glomerulonephritis and urine protein-to-creatine ratios (UPCRs) > 1 g/g who were treated with 25 mg of spironolactone on top of ACEi or ARBs vs. a matched placebo on top of ACEi or ARBs. These authors demonstrated a reduction in UPCR at 1 year from 2.1 to 0.9 g/g. 

Eplerenone, has been studied in multiple CKD settings, diabetic and not. The EVALUATE trial randomized 314 non-diabetic CKD patients with UACRs of 30–599 mg/g to receive 50 mg/day eplerenone or a placebo [114]. After 1 year of follow up, the primary endpoint (percent reduction in UACR) was significantly lower in the eplerenone group vs. the placebo (−17.3% vs. −10.3%; *p* = 0.02).

New-generation MRAs finerenone and exarenone are non-steroidal MRAs that, given their more bulky molecular structure, have enhanced receptor selectivity, strong MR inhibition, and a shorter half-life with no active metabolites, which may be associated with less frequent hyperkaliemia, thereby increasing their safety profile [115,116,117]. 

Sound evidence supporting the use of new MRAs comes from recent clinical trials. FIGARO-DKD enrolled a total of 7437 T2DM patients with kidney disease. The primary outcome was a composite of death from cardiovascular causes, nonfatal myocardial infarction, nonfatal stroke, or hospitalization for heart failure. The first predefined secondary outcome was a composite of kidney failure, a sustained decrease of at least 40% from the baseline in the eGFR, or death from renal causes. The renal outcome was less frequent in the finerenone group but failed to achieve formal statistical significance (HR 0.87; 95% CI = 0.76–1.01; *p* = 0.07) [118].

The protective effect of finerenone on primary kidney endpoints was confirmed in the FIDELIO trial (Table 1) [119]. The study randomized 5674 patients with DKD in a 1:1 ratio to receive finerenone or a placebo. Included patients had UACRs of 30 to less than 300 mg/g, an eGFR of 25 to less than 60 mL/min/1.73 m^2^, and diabetic retinopathy, or they had a UACR of 300 to 5000 mg/g and an eGFR of 25 to less than 75 mL/min/1.73 m^2^. Notably, all the patients were treated with RAS blockade. The two parallel groups were followed for a median of 2.6 years. The hazard ratio (HR) of the primary composite renal outcome was significantly reduced by finerenone (HR 0.82; 95% CI 0.73–0.93; *p* = 0.001) [119].

Pooling the data of FIGARO [118] and FIDELIO [119], FIDELITY [120] analyzed the renal and cardiovascular outcomes in the finerenone trials. The analysis showed a highly significant beneficial effect on kidney outcomes (HR = 0.77; 95% CI 0.67–0.88; *p* = 0.0002), with a number needed to treat (NNT) of 60 patients after 3 years (95% CI 38–142). 

### 3.4. Hypoxia-Inducible Factor

HIF pleiotropic activity across different metabolisms and the relative weakness of clinical evidence for HIF-PHI impose great caution for the use of inhibitors/activators of this pathway in the prevention of renal fibrosis and CKD progression. 

Roxadustat was the first HIF-PHI approved for renal anemia in China in 2019. Shortly after, Japan approved five HIF-PHIs, including roxadustat, daprodustat, vadadustat, molidustat, and enarodustat, for the same indication. However, in the European Union, only roxadustat is currently available for the treatment of adult patients with symptomatic anemia associated with CKD [121]. 

The evidence supporting the roxadustat indication in renal anemia is mainly derived from phase III clinical trials conducted in both non-dialysis CKD and dialysis-dependent CKD patients [122,123,124,125]. Unfortunately, none of these studies had CKD progression or fibrosis as a prespecified outcomes. Hence, at the moment, the clinical efficacy of HIF-PHI compounds on renal fibrosis and CKD progression remains unproven.

**Table 1 biomolecules-14-00137-t001:** Kidney disease-focused primary-outcome trials with sodium-glucose cotransporter-2 inhibitors, glucagon-like peptide-1 receptor agonists, and non-steroidal mineralocorticoid receptor antagonists.

Trial, Year	Design	Intervention	Class	Number of Patients	Follow Up, Years	Stopped Early for Benefit	Reference
CREDENCE, 2019	RCT	Canagliflozin vs. placebo	SGLT2-i	4401	2.6	Yes	[95]
DAPA-CKD, 2020	RCT	Dapagliflozin vs. placebo	SGLT2-i	4304	2.4	Yes	[96,97]
EMPA-KIDNEY, 2023	RCT	Empagliflozin vs. placebo	SGLT2-i	6609	2.0	Yes	[98]
FLOW, 2024	RCT	Semaglutide vs. placebo	GLP-1 RAs	3534	3–5 (expected)	Yes	[105,106]
FIDELIO-DKD, 2020	RCT	Finerenone vs. placebo	NS-MRAs	5734	2.6	No	[119]

RCT, randomized controlled trial; CKD, chronic kidney disease; DKD, diabetic kidney disease; SGLT2-I, sodium-glucose cotransporter-2 inhibitors; GLP-1 RAs, glucagon-like peptide-1 receptor agonists; NS-MRAs, non-steroidal mineralocorticoid receptor antagonists.

## 4. Conclusions

The molecular basis and the fundamental aspects supporting a potentially causal association between kidney fibrosis and oxidative stress are well documented in cellular and animal model studies. However, despite sound and consistent evidence on clinical outcomes, there is a distinct paucity of data on the efficacy of new drugs on kidney fibrosis in humans. Hence, there is still a long and winding road to a clear understanding of the anti-fibrotic effects of these compounds in humans, perhaps due to the inherent practical and ethical difficulties in obtaining sequential kidney biopsies and the lack of accurate fibrosis biomarkers measurable in easily accessible matrices like urine.

## Figures and Tables

**Figure 1 biomolecules-14-00137-f001:**
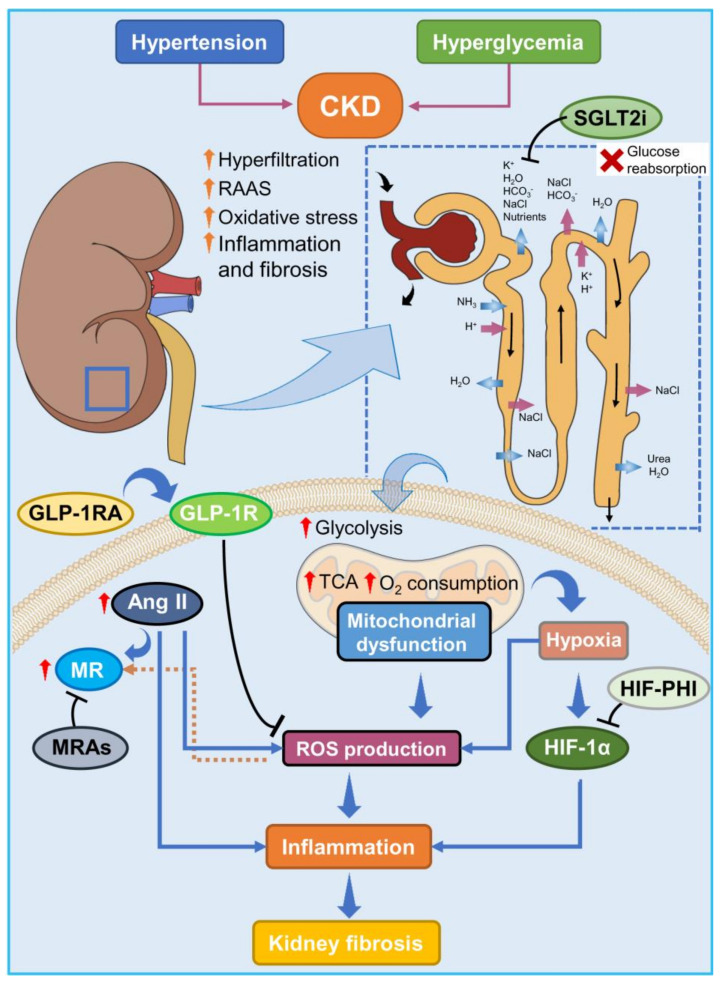
Pathomechanisms driving chronic kidney disease and novel disease-modifying treatment strategies. Hypertension and hyperglycemia are the main causes of CKD, leading to progressive kidney damage, as depicted by the formation of fibrotic tissue. Abnormal activity of RAAS triggers oxidative stress, inflammatory pathways, and fibrinogenesis. Under hyperglycemic conditions, proximal tubule cells have enhanced energy requirements to sustain the incessant reuptake of glucose through the SGLT2 transporters, resulting in altered mitochondrial activity and ROS generation and finally triggering the hypoxia pathway. Many therapeutic strategies under development aim to ameliorate oxidative stress, to dampen the associated inflammatory response, and to slow down the fibrotic tissue deposition with the use of drugs targeting different pathways, such as MR antagonists, GLP-1R agonists, SGLT2 transporter inhibitors, and HIF-PHD inhibitors. Ang II, angiotensin II; CKD, chronic kidney disease; GLP-1R, glucagon-like peptide 1 receptor; GLP-1RA, glucagon-like peptide 1 receptor agonist; HIF-1α, hypoxia-inducible factor 1α; HIF-PHI, HIF-PHD inhibitor; MR, mineralocorticoid receptor; MRAs, mineralocorticoid receptor antagonists. RAAS, renin–angiotensin–aldosterone system; ROS, reactive oxygen species; SGLT2i, sodium glucose transporter 2-inhibitors; TCA, tricarboxylic acid cycle.

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
