# Peer review of "Kidney Fibrosis and Oxidative Stress: From Molecular Pathways to New Pharmacological Opportunities"

_biomolecules, 2024, doi:10.3390/biom14010137_

Round 1

Reviewer 1 Report

Comments and Suggestions for Authors

This is a timely review.

Some sugestions.

1) English language editiong will be very beneficial.

2) The renal protective roles of SLP-1 Receptor agonists should be discussed.

3) AQP2 is not a proximal tubule protein marker.

4) The review would highly benefit from a schematic/mechanistic summary figure.

Comments on the Quality of English Language

English editing will be highly beneficial: starting with the first few sentences in the Introduction.

Author Response

We appreciate the reviewers’ valuable and insightful comments. The authors have carefully considered the comments and tried the best to address every one of them.

Reviewer 2 Report

Comments and Suggestions for Authors

Review of manuscript biomolecules-2748935

Kidney fibrosis and oxidative stress: from molecular pathways to new pharmacological opportunities

by Francesco Patera et al.

The evaluated manuscript is a narrative review. The authors focus on the description of pathomechanisms contributing to the development of increased oxidative stress and fibrosis in patients with chronic kidney disease (CKD). After a short Introduction, Authors discuss in detail three pathways inducing oxidative stress and renal fibrosis: hyperglycemic-, mineralocorticoid- and hypoxia-induced factor (HIF)-related mechanisms at the molecular level. The next part of the article ("Clinical scenario") briefly discusses clinical trials demonstrating the use of selected compounds: SGLT2 inhibitors, aldosterone antagonists and compounds that affect HIF action as potential drugs reducing the progression of kidney damage in CKD.

The manuscript is well-thought-out, a synthetic description of the issues mentioned above and it may have educational value also for non-nephrologists.

However, I think that before possible publication, Authors may consider suggestions for improving the manuscript:

1. There are a significant number of abbreviations throughout the manuscript. I believe that the reading of the text would be made easier by introducing a list of abbreviations used in the text

2. The authors focus on the 3 main pathomechanisms of oxidative stress and renal fibrosis mentioned above. However, a wider outline of a more comprehensive description of these disturbances should be considered, also mentioning potential laboratory markers.

3. Similarly, in the article, Authors describe in details three classes of drugs in terms of their use in preventing the progression and complications of CKD resulting from oxidative stress and fibrosis. However, it is also worth mentioning, at least in the margin of the main considerations, other compounds evaluated for their potential effectiveness in reducing oxidative stress and renal fibrosis, such as antioxidants, xanthine oxidase inhibitors, TGF-beta inhibitors, compounds acting through Nrf2, etc.

For example, based on:

https://www.termedia.pl/Oxidative-stress-mechanisms-as-potential-therapeutic-targets-in-chronic-kidney-disease,67,47380,1,1.html

https://www.mdpi.com/2076-3921/9/8/752

https://www.mdpi.com/2076-3921/10/2/258

4. The Authors mention the importance of nephroprotective drugs that inhibit the activity of the RAA system in reducing oxidative stress and renal fibrosis (e.g. lines 69-78). There is a question, if there are any scientific evidence for the possible use of another group of compounds - renin inhibitors – also as nephroprotective drugs?

5. The Authors discuss in detail the role of SGLT2 in CKD. Similarly to the abovementioned remark, is there scientific evidence showing a beneficial effect on the course of oxidative stress and renal fibrosis in patients with diabetes and CKD, but treated with other hypoglycemic drugs, e.g. biguanides or sulfonylureas?

6. In the "Clinical scenario" chapter, the Authors briefly describe the clinical trials focusing on the effectiveness of the selected drugs discussed in the manuscript. I would suggest introducing a collective table synthetically summarizing the content provided in this chapter. The table could contain columns, e.g. authors, year of publication, type of study, studied arms with group size, inclusion criteria, assessed primary/secondary end points, results with statistical significance

Author Response

(The authors gave the same response as above.)
